

# How SU(2)$_4$ anyons are $\mathbb{Z}_3$ parafermions

**Richard Fern[1], Johannes Kombe[2] and Steven H. Simon[1]**

**1** Rudolf Peierls Centre for Theoretical Physics, Oxford OX1 3NP, United Kingdom
**2** HISKP, University of Bonn, Nussallee 14-16, 53115 Bonn, Germany

## Abstract

We consider the braid group representation which describes the non-abelian braiding statistics of the spin $1/2$ particle world lines of an SU(2)$_4$ Chern-Simons theory. Up to an abelian phase, this is the same as the non-Abelian statistics of the elementary quasiparticles of the $k = 4$ Read-Rezayi quantum Hall state. We show that these braiding properties can be represented exactly using $\mathbb{Z}_3$ parafermion operators.


## 1 Motivation

The proposition of employing topological matter to perform fault-tolerant quantum computation is a truly exciting one [1, 2]. Within this framework quantum gates are executed by *braiding* (exchanging) the quasiparticle excitations of $(2 + 1)$D systems. If the ground state of a system is degenerate then these operations can act as unitary matrices on the degenerate

Hilbert space. If these matrices, which form a representation of the *braid group*, are then sufficiently rich that any unitary transformation can be generated then one could, in principle, use it to create a *universal* quantum computer [1–4].

The key advantage of the topological scheme lies in its fault tolerance. The exchange of any two quasiparticles, usually referred to as *anyons*, depends only on the topology of the paths traversed by the particles and this protects computations from errors due to local noise, which are difficult to eradicate in alternative proposals.

Systems containing particles with nontrivial exchange statistics are described by Topological Quantum Field Theories (TQFTs) [2, 5–9]. In this work we focus on TQFTs related to $SU(2)_k$ Chern-Simons TQFTs for the specific case of $k = 4$. The braiding statistics of the spin-1/2 particles from this theory precisely describe the braiding of the elementary quasiparticles of the $k = 4$ Read-Rezayi quantum Hall state [10] of bosons [11] at filling fraction 2. The fermionic version of this quantum Hall state (whose braiding properties differ from that of the bosonic version, which is pure $SU(2)_4$, by only a trivial abelian phase) describes electrons in a 2/3 filled Landau level.[1] There is some recent numerical evidence that the experimentally observed $\nu = 8/3$ fractional quantum Hall state is of this type [12].

It was previously believed that the braid group representation corresponding to the $SU(2)_4$ type anyons was (like that of $SU(2)_2$ or Ising anyons) not sufficiently rich to allow for universal quantum computation [13]. However, recent work [14–16] has shown that, by including fusion and measurement operations, such a system can in fact be made universal [14–16]. This realisation has greatly increased interest in this type of anyon.

At the same time, there has also been increasing interest in so-called $\mathbb{Z}_k$ parafermionic systems of Fradkin-Kadanoff-Fendley type [17–30]. Such parafermions were introduced by Fradkin and Kadanoff [17] as mathematical operators which allowed for elegant solutions of certain two dimensional statistical models with $\mathbb{Z}_k$ symmetry.

More recently it was realised that these same mathematical operators may be used to describe defects that can arise in certain topological models [18, 21–31] — and there have been several proposals to experimentally engineer such defects. In this context the defects can be thought of as particles having some variety of non-abelian braiding statistics.

We caution the reader that the level-$k$ Read-Rezayi wavefunction [10] is constructed from a conformal field theory (CFT) known as the $\mathbb{Z}_k$ parafermions of Zamolodchikov-Fateev [32] type which is closely related to the $SU(2)_k$ Chern-Simons theory, and which is a different object from the parafermions of the Fradkin-Kadanoff-Fendley type.[2]

To avoid confusion we emphasise at this point that within this paper, all mention of parafermions will refer to Fradkin-Kadanoff-Fendley type — that is, they are mathematical operators (to be defined precisely in Eqs. 19 and 20 below).

The content of the current work is to express the braid group representation corresponding to $SU(2)_4$ anyons explicitly in terms of $\mathbb{Z}_3$ parafermion operators. This relationship is analogous to the one between the $SU(2)_2$ anyons and Majorana fermion operators — which are otherwise known as $\mathbb{Z}_2$ parafermions [20, 33].

It should be noted that links between these two different mathematical structures are implied in several prior works [21, 23, 30, 34–39]. However, in these works the relationship between the two braid group representations is never made explicit. Further, our step forward of expressing a non-abelian braid group representation in terms of a simple operator algebra is a very powerful advance in our understanding. For example, we know that the braid matrices

---

[1]For simplicity we only consider two versions of the $k = 4$ Read-Rezayi state, $\nu = 2$ for bosons and $\nu = 2/3$ for fermions. In general attaching $M$ Jastrow factors to the $SU(2)_4$ wavefunction gives a trial wavefunction for $\nu = 2/(1 + 2M)$, which is bosonic for $M$ even and fermionic for $M$ odd. One then uses a value of $\alpha = \exp(\pi i \nu[1 - M]/24)$ for the phase in the braiding relations, 4-6. [10, 11]

[2]In fact the Zamolodchikov-Fateev parafermions represent a critical point at the transition into a phase described by Fradkin-Kadanoff-Fendley parafermions.

of Ising (or SU(2)$_2$) anyons can be expressed in terms of simple Majorana operators [40–42]. This structure, in turn, not only helps us devise computational algorithms [43] but also helps us understand the fundamental (in the Majorana case, fermionic) structure of the underlying physics. The simple operator representation we obtain for the SU(2)$_4$ anyons should be similarly useful — and potentially even more exciting, given that this system is capable of universal quantum computation.

## 2  Theoretical background

**Braid group** — Let us line up $N$ particles living in two dimensions along a one dimensional line. The elementary braid group generators $b_i$ exchange clockwise the $i^{th}$ and $(i+1)^{th}$ particles. The braid group on $N$ particles [2,3] is generated by the $b_i$'s for $i = 1 \dots N-1$ as well as their inverses, and supplemented by the relations

$$b_i b_j = b_j b_i \qquad \text{for} \qquad |i-j| > 1, \tag{1}$$
$$b_i b_{i+1} b_i = b_{i+1} b_i b_{i+1}. \tag{2}$$

A system of $N$ anyons in 2D must correspond to some representation of this braid group where each braid group element $b_i$ is represented by a corresponding matrix $\mathscr{B}_i$.

**SU(2)$_4$ anyons** — The TQFT corresponding to the group SU(2)$_k$ is a theory containing $k+1$ particle types which fuse together in much the same way as spins add, but with the caveat that the total spin of any composite cannot exceed some maximum [2,11,44]. We label these particles by some spin, $n/2$ where $n$ runs from 0 to $k$, and in this way the maximum spin is $k/2$. The spin-1/2 particles are then the fundamental particles of the theory and fusing many such particles allows us to generate any of the other $k$ particle types.

The *Bratteli diagram* in Figure 1, represents all possible fusions of $N$ spin-1/2 particles as paths through intermediate configurations for the case of SU(2)$_4$ [2,44]. Each different path on this diagram represents a different state in the $N$-particle Hilbert space. For example, two spin-1/2 particles might fuse to a total spin of 1 or 0. Upon adding a third particle, the former configuration could generate either a spin-1/2 or a spin-3/2 particle whereas the latter will always yield a spin-1/2. Therefore, the 3-particle system is made up of the states

$$\left| \overset{}{\underset{\sim}{=}} \right\rangle, \left| \overset{}{\underset{}{\diagup\!\diagdown}} \right\rangle \text{ and } \left| \overset{}{\underset{}{\diagup}} \right\rangle. \tag{3}$$

Let us now consider the braid group representation for these SU(2)$_4$ anyons. The detailed structure of this representation has been worked out explicitly by Slingerland and Bais [11]. While the derivation of their result is somewhat complicated, the results are fairly easy to interpret. The braid matrix $\mathscr{B}_i$, which clockwise exchanges the $i^{th}$ particle in the sequence with the $(i+1)^{th}$ particle, can only alter the intermediate fusion of the $i$ particles (i.e, the spin value at the $i^{th}$ step of the diagram). The reason for this is that the first $(i-1)$ anyons are not moved, so their fusion at the $(i-1)^{th}$ step is unchanged. Further, viewing the group of $(i+1)$ particles from far away, by locality, its overall spin is not changed by braiding particles within the group. Thus only the spin value at the $i^{th}$ step may be changed.

As a result of this principle the exchange of two successive up-steps or two successive down-steps in the Bratteli diagram can only produce a phase. The phase was found by [11] to be the same for all up-up and down-down diagrams, with

$$\mathscr{B}_i \left| \overset{}{\underset{}{\diagup}} \right\rangle = \alpha \left| \overset{}{\underset{}{\diagup}} \right\rangle, \qquad \mathscr{B}_i \left| \overset{}{\underset{}{\diagdown}} \right\rangle = \alpha \left| \overset{}{\underset{}{\diagdown}} \right\rangle, \tag{4}$$

where $\alpha$ is defined below. The height of these segments (i.e, the spins they begin from) are unimportant to this relation.

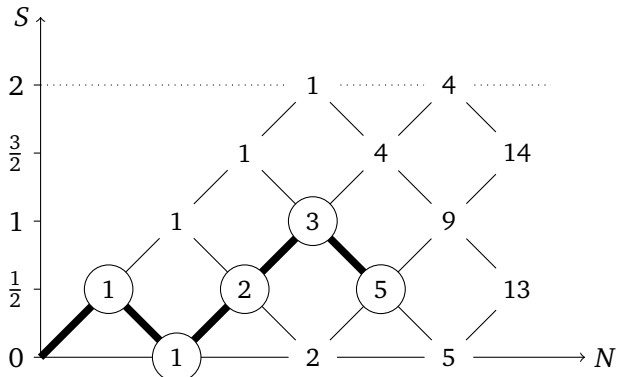

Figure 1: This *Bratteli diagram* shows the various possible states in the Hilbert space of $N$ spin-$\frac{1}{2}$ particles in SU(2)$_4$. Each path segment represents the addition of one spin-$\frac{1}{2}$ particle. The numbers at the vertices denote the dimension of the Hilbert space at that point (i.e, the number of ways that $N$ particles can fuse to spin $S$). The highlighted path is an example demonstrating one way in which 5 particles might fuse to a total spin of $\frac{1}{2}$.

When one considers an up-down sequence or a down-up sequence in the Bratteli diagram, the situation is more complex and the action of the braid matrices is as follows:

$$\mathscr{B}_i \left| \overset{\frown}{\overline{\phantom{x}}} \right\rangle = \alpha \frac{\omega^{k-2S}}{\lfloor 2S+1 \rfloor} \left| \overset{\frown}{\overline{\phantom{x}}} \right\rangle - \alpha \frac{\sqrt{\lfloor 2S \rfloor \lfloor 2S+2 \rfloor}}{\omega \lfloor 2S+1 \rfloor} \left| \underset{\smile}{\overline{\phantom{x}}} \right\rangle, \tag{5}$$

$$\mathscr{B}_i \left| \underset{\smile}{\overline{\phantom{x}}} \right\rangle = \alpha \frac{\omega^{2S}}{\lfloor 2S+1 \rfloor} \left| \underset{\smile}{\overline{\phantom{x}}} \right\rangle - \alpha \frac{\sqrt{\lfloor 2S \rfloor \lfloor 2S+2 \rfloor}}{\omega \lfloor 2S+1 \rfloor} \left| \overset{\frown}{\overline{\phantom{x}}} \right\rangle, \tag{6}$$

where the centre (vertical position) of these diagrams are assumed to be at spin $S$ and we define $\omega$ and $\lfloor x \rfloor$ below. Therefore, the exchange of two SU(2)$_4$ anyons either leaves the intermediate quantum number unchanged or changes the spin by 1.

In the above relations $\omega = \exp(i\pi/6)$ and the phase $\alpha = \exp(i\pi/12) = \sqrt{\omega}$ for pure SU(2)$_4$. For the fermionic version of the quantum Hall wavefunction (corresponding to a 2/3 filled Landau level), we instead use $\alpha = 1$. Finally, the function $\lfloor x \rfloor$ is given by $\lfloor x \rfloor = 2\sin(\pi x/6)$.

All possible actions of the braid operator are listed explicitly in table 1 for convenience and later reference.

$\mathbb{Z}_3$ **parafermions** — The $\mathbb{Z}_n$ clock model, a generalisation of the Ising chain, can be mapped to a model of free parafermions. This is exactly analogous to the mapping between the Ising model and a free Majorana fermion system in one dimension [20]. In fact, Majoranas can be thought of as $\mathbb{Z}_2$ parafermions.

The Hilbert space of the clock model is one of a chain of clocks with $n$ states per face, denoted by $|l\rangle$ for $l \in \mathbb{Z}_n$. This is represented pictorially in figure 2 for $n = 3$. We then have two operators per site: $\sigma$, which measures the clock value, $\sigma |l\rangle = \lambda^l |l\rangle$ where $\lambda = \exp(2\pi i/n)$; and $\tau$, which increments the clock, $\tau |l\rangle = |l+1\rangle$ where $l$ is understood to be defined modulo $n$. These operators satisfy the algebra $\sigma\tau = \lambda\tau\sigma$ on a given site, and commute on different sites.

The simplest generalisation of the Ising Hamiltonian is the following,

$$H = \sum_{j=1}^{N} t_{2j-1} \tau_j + \sum_{j=1}^{N-1} t_{2j} \sigma_j^\dagger \sigma_{j+1}, \tag{17}$$

where these couplings, $t_j$, are real. The first term in this Hamiltonian generalises the flip due to a transverse magnetic field and the second term is a nearest-neighbour interaction.

$$\mathcal{B}_i \left| \overset{\frown}{\underset{=}{}} \right\rangle = \frac{\alpha\omega}{\sqrt{3}} \left| \overset{\frown}{\underset{=}{}} \right\rangle - \frac{\alpha\sqrt{2}}{\omega\sqrt{3}} \left| \underset{\smile}{=} \right\rangle \tag{7}$$

$$\mathcal{B}_i \left| \underset{\smile}{=} \right\rangle = \frac{\alpha\omega}{\sqrt{3}} \left| \underset{\smile}{=} \right\rangle - \frac{\alpha\sqrt{2}}{\omega\sqrt{3}} \left| \overset{\frown}{\underset{=}{}} \right\rangle \tag{8}$$

$$\mathcal{B}_i \left| \overset{\frown}{=} \right\rangle = \frac{\alpha\omega^3}{\sqrt{3}} \left| \overset{\frown}{=} \right\rangle - \frac{\alpha\sqrt{2}}{\omega\sqrt{3}} \left| \underset{\smile}{=} \right\rangle \tag{9}$$

$$\mathcal{B}_i \left| \underset{\smile}{=} \right\rangle = \frac{\alpha\omega^3}{\sqrt{3}} \left| \underset{\smile}{=} \right\rangle - \frac{\alpha\sqrt{2}}{\omega\sqrt{3}} \left| \overset{\frown}{=} \right\rangle \tag{10}$$

$$\mathcal{B}_i \left| \diagup \right\rangle = \alpha \left| \diagup \right\rangle \qquad \mathcal{B}_i \left| \diagdown \right\rangle = \alpha \left| \diagdown \right\rangle \tag{11}$$

$$\mathcal{B}_i \left| \underset{\smile}{\overset{\smile}{}} \right\rangle = \alpha\omega^4 \left| \underset{\smile}{\overset{\smile}{}} \right\rangle \qquad \mathcal{B}_i \left| \overset{\frown}{\underset{\frown}{}} \right\rangle = \alpha\omega^4 \left| \overset{\frown}{\underset{\frown}{}} \right\rangle \tag{12}$$

$$\mathcal{B}_i \left| \diagup \right\rangle = \alpha \left| \diagup \right\rangle \qquad \mathcal{B}_i \left| \diagup \right\rangle = \alpha \left| \diagup \right\rangle \tag{13}$$

$$\mathcal{B}_i \left| \diagdown \right\rangle = \alpha \left| \diagdown \right\rangle \qquad \mathcal{B}_i \left| \diagdown \right\rangle = \alpha \left| \diagdown \right\rangle \tag{14}$$

$$\mathcal{B}_i \left| \overset{\frown}{=} \right\rangle = \frac{\alpha\omega^2}{2} \left| \overset{\frown}{=} \right\rangle - \frac{\alpha\sqrt{3}}{2\omega} \left| \underset{\smile}{=} \right\rangle \tag{15}$$

$$\mathcal{B}_i \left| \underset{\smile}{=} \right\rangle = \frac{\alpha\omega^2}{2} \left| \underset{\smile}{=} \right\rangle - \frac{\alpha\sqrt{3}}{2\omega} \left| \overset{\frown}{=} \right\rangle \tag{16}$$

Table 1: The braid operator for $SU(2)_4$ anyons applied to all possible Bratteli basis states. For pure $SU(2)_4$ one uses $\alpha = \exp(i\pi/12)$, whereas for the corresponding fermionic k=4 Read-Rezayi quantum Hall state describing a 2/3 filled Landau level, one uses $\alpha = 1$.

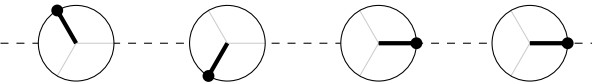

Figure 2: Each site of the $\mathbb{Z}_3$ clock model is a three-state system which we can visualise as a clock pointing in a given direction. Shown is the $|\dots, 1, 2, 0, 0, \dots\rangle$ state.

This Hamiltonian is equivalent to a system of free parafermions hopping on a chain. The mapping between the two is via the Fradkin-Kadanoff transformation [17], a non-local transformation which creates two parafermionic degrees of freedom per site $j$ via

$$\psi_{2j-1} = \left( \prod_{m=1}^{j-1} \tau_m \right) \sigma_j, \qquad \psi_{2j} = \lambda^{\frac{n-1}{2}} \psi_{2j-1} \tau_j. \tag{18}$$

For the $n = 2$ case of Majoranas, these operators simply anti-commute and square to unity as expected. For general $n$, the algebra is as follows:

$$\left( \psi_j \right)^n = 1, \qquad \psi_j^\dagger = \left( \psi_j \right)^{n-1}, \tag{19}$$

$$\psi_j \psi_{j+m} = \lambda \psi_{j+m} \psi_j, \tag{20}$$

where $m > 0$. In this language, the Hamiltonian takes the form

$$H = \lambda^{\frac{n-1}{2}} \sum_{j=1}^{2N-1} t_j \psi_{j+1} \psi_j^\dagger. \tag{21}$$

Therefore, in the same way that the Ising chain can be thought of as a one-dimensional wire of free Majorana modes, the $\mathbb{Z}_n$ clock chain's free modes are parafermions.

In several recent works, it has been shown how physical systems can be constructed to have free $\mathbb{Z}_n$ parafermion modes [18–23, 27]. In these models, braiding matrices can generally be written in the form [28, 29]

$$B_i = \frac{1}{\sqrt{n}} \sum_{k=0}^{n-1} c_k (\psi_i \psi_{i+1}^\dagger)^k \tag{22}$$

for certain complex coefficients $c_k$. There are several possible sets of values of the coefficients $c_k$ which can be strongly constrained by forcing these matrices to be both unitary and a representation of the braid group.

This parafermion braid representation generalises the well known braiding matrices for $n = 2$ Majorana case of the form [40–42]

$$B_i = \frac{1}{\sqrt{2}} (1 + \psi_i \psi_{i+1}), \tag{23}$$

where $\psi = \psi^\dagger$.

Although the local degrees of freedom are most simply described in terms of these parafermionic operators, in what follows we will predominantly work in the clock basis, $\sigma$ and $\tau$, as this will simplify calculations. However, it is relatively straightforward to convert these operators into parafermions via, for the $n = 3$ case,

$$\tau_j = \lambda^{-1} \psi_{2j-1}^\dagger \psi_{2j}, \tag{24}$$

$$\sigma_j = \left( \prod_{m=1}^{j-1} \tau_m^\dagger \right) \psi_{2j-1}. \tag{25}$$

## 3 Result

We claim that the braid group representation of $SU(2)_4$ anyons can be represented by the operators

$$B_j = \frac{\alpha \omega}{\sqrt{3}} \left( 1 + \psi_j^\dagger \psi_{j+1} + \psi_j \psi_{j+1}^\dagger \right). \tag{26}$$

In what follows however, the picture is clearer when working in the clock basis. In this language the matrices have the following form

$$B_{2j-1} = \frac{\alpha \omega}{\sqrt{3}} \left( 1 + \omega^4 \left( \tau_j + \tau_j^\dagger \right) \right), \tag{27}$$

$$B_{2j} = \frac{\alpha \omega}{\sqrt{3}} \left( 1 + \omega^4 \left( \sigma_j^\dagger \sigma_{j+1} + \sigma_j \sigma_{j+1}^\dagger \right) \right). \tag{28}$$

With some algebra it is easy to establish that these matrices satisfy the braid relations Eqs. 1 and 2. These calculations are demonstrated in appendix A.

To clarify the connection between the parafermions and the $SU(2)_4$ anyons, we will demonstrate the explicit connection between states in the $SU(2)_4$ Bratteli diagram, and states as described in the (parafermion) clock model. Each parafermion represents one anyon so each clock represents two anyons.

In what follows we restrict our Hilbert space to systems of $2N$ anyons with total spin 0 or $k/2$ without loss of generality (as any state can be represented by a $2N$-particle state with these restrictions via the addition of at most 2 superfluous anyons). Furthermore, we will work in the Fourier basis for the parafermions. For each clock site, we define

$$|\tilde{l}\rangle = \sum_{p=0}^{2} \lambda^{-lp} |p\rangle \tag{29}$$

so that $\tau|\tilde{l}\rangle = \lambda^l |\tilde{l}\rangle$ and $\sigma|\tilde{l}\rangle = \widetilde{|l-1\rangle}$ and $l$ is defined modulo 3 (one may wonder why $\sigma$ now decrements the clock where previously $\tau$ incremented it but this follows from the algebra of the operators which, given $\tau\sigma|\tilde{l}\rangle = \lambda^{l-1}\sigma|\tilde{l}\rangle$, implies that $\sigma|\tilde{l}\rangle = \widetilde{|l-1\rangle}$).

We begin by defining the vacuum state for the parafermions as the Bratteli diagram where each successive pair of anyons fuse to spin zero:

$$|\Phi_N\rangle = \overbrace{|\tilde{0}\rangle_1 \otimes |\tilde{0}\rangle_2 \otimes \ldots \otimes |\tilde{0}\rangle_N}^{N \text{ clock sites}}$$
$$= \left|\begin{array}{c} \rule{1cm}{0pt} \end{array}\right\rangle \tag{30}$$

where the subscript on $|\tilde{0}\rangle_j$ indicates that this is the value of the $j^{\text{th}}$ clock.

The Hilbert space for the anyons is then generated by the operators

$$P_j^{\pm} = \frac{1}{\omega^4\sqrt{2}}\left(\psi_{2j}^{\dagger}\psi_{2j+1} \pm \psi_{2j}\psi_{2j+1}^{\dagger}\right) \tag{31}$$

and the identity $\mathbb{1}_j$. It is crucial to note that all of these operators commute with each other. We claim that the action of these operators on $|\Phi_N\rangle$ generates the following path segments in the anyonic Hilbert space

$$\mathbb{1}_j|\Phi_N\rangle \equiv \left|\begin{array}{c}\end{array}\right\rangle \text{ or } \left|\begin{array}{c}\end{array}\right\rangle, \tag{32}$$

$$P_j^+|\Phi_N\rangle \equiv \left|\begin{array}{c}\end{array}\right\rangle \text{ or } \left|\begin{array}{c}\end{array}\right\rangle, \tag{33}$$

$$P_j^-|\Phi_N\rangle \equiv \left|\begin{array}{c}\end{array}\right\rangle \text{ or } \left|\begin{array}{c}\end{array}\right\rangle, \tag{34}$$

where these ambiguities are resolved by noting whether the previous $j-1$ path operators produce a Bratelli state of total spin 1/2 or 3/2. In the clock basis this translates to an even or odd number of $P_j^-$ operators respectively. Once again, these states can be phrased in terms of the clock basis as

$$P_j^{\pm} = \frac{1}{\sqrt{2}}\left(\sigma_j^{\dagger}\sigma_{j+1} \pm \sigma_j\sigma_{j+1}^{\dagger}\right) \tag{35}$$

and are orthogonal and normalised, as we will demonstrate later.

The physical picture to keep in mind is then one of the clock model sites sitting at the odd particle numbers on the Bratteli diagram with these *path operators*, $\mathbb{1}_j$ and $P_j^{\pm}$, living on the bonds of the clock chain. Each clock degree of freedom is acted on by the two adjacent bonds. This is demonstrated by figure 3, which shows the string of operators required to reproduce the Bratteli diagram shown.

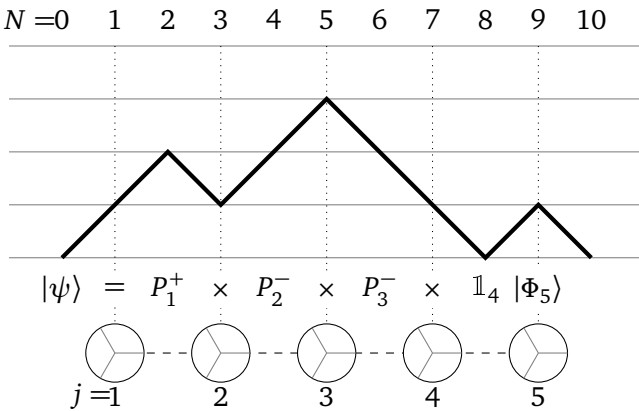

Figure 3: A diagram showing a representation of a particular Bratteli state in terms of the parafermionic operators. The path operators create two-leg segments of the Bratteli path and live on the bonds of the clock model.

It should also be stressed that this formulation of the braid group representation of $SU(2)_4$ anyons in terms of $\mathbb{Z}_3$ parafermions is not unique. It is, however, the simplest formulation in which the braiding matrices can be phrased in terms of the parafermions in some local way (i.e, the braid matrix $B_j$ depends only on $\psi_j$ and $\psi_{j+1}$ and not all the $\psi_i$ for $i < j$ as well). Further discussion on these alternative formulations is presented in appendix B.

Thus we have defined the Braid matrices in the parafermionic language via Eq. 26 (or equivalently Eqs. 27 and 28) and we have defined the mapping of the clock states in the parafermionic language to the Bratteli states of $SU(2)_4$. In the next section we will prove the equivalence of the braid matrices in the two languages (i.e., the equivalence of Eqs. 27 and 28 with the action of the braid matrices for $SU(2)_4$ listed in table 1). In addition, in appendix B we give additional motivation for why the mapping between the two descriptions should be exactly as we have described. This should hopefully make the mapping seem a bit less mysterious.

## 4  Proof

It is important that our proposed basis of states is a suitable one with which to represent the $SU(2)_4$ Hilbert space — i.e, that it have the right dimension. For $SU(2)_4$ the dimension of a system of $2N$ anyons restricted to a total spin of 0 or $k/2$ is $3^{N-1}$. This is because the first segment of any Bratteli path is simply $\left| \begin{smallmatrix} - \\ \diagup \end{smallmatrix} \right\rangle$ and every subsequent segment (of two steps) can only be of the form $\mathbb{1}_j$, $P_j^+$, or $P_j^-$ as graphically indicated above. The final step can then always be chosen as either $\left| \begin{smallmatrix} - \\ \diagdown \end{smallmatrix} \right\rangle$ or $\left| \begin{smallmatrix} \diagup \\ - \end{smallmatrix} \right\rangle$. Thus the total number of paths is $3^{N-1}$. Similarly, it is easy to see that $\mathbb{1}_j$, $P_j^+$ and $P_j^-$ produce three linearly independent states per bond on the $N$-site clock chain. Thus our parafermionic basis does indeed have dimension $3^{N-1}$.

We note in passing that the Hilbert space dimension of $2N$ parafermion operators is of the form $M^N$ for $\mathbb{Z}_M$ parafermions. Amongst the theories $SU(2)_k$, only $k = 1, 2, 4$ have Hilbert space dimensions of this form ($M = 1, 2, 3$ for $k = 1, 2, 4$ respectively). The case $k = 1$ is a trivial abelian theory, $k = 2$, as mentioned above, is related to the Ising or Majorana case, and $k = 4$ is addressed here. Because matching of Hilbert space dimension is necessary to represent braiding operations, no other value of $k$ lends itself to a similar mapping to parafermion operators.

Our proposed clock-basis states are also orthogonal and normalised like those in the $SU(2)_4$ Hilbert space. To see this we shall for the moment denote the trivial path operator $\mathbb{1}_j = P_j^0$. Therefore, a general state can be indexed by a set $\boldsymbol{\alpha} = \{\alpha_1, \ldots \alpha_{N-1}\}$ where each $\alpha_j \in \{-, 0, +\}$ as defined by

$$|\boldsymbol{\alpha}\rangle = \prod_{j=1}^{N-1} P_j^{\alpha_j} |\Phi_N\rangle. \tag{36}$$

Given that all $P_j^{\alpha_j}$ commute, the inner product of any two states $|\boldsymbol{\alpha}\rangle$ and $|\boldsymbol{\beta}\rangle$ has the form

$$\langle \boldsymbol{\alpha} | \boldsymbol{\beta} \rangle = \langle \Phi_N | \prod_j \left( \left( P_j^{\alpha_j} \right)^\dagger P_j^{\beta_j} \right) |\Phi_N\rangle. \tag{37}$$

It is easy to check by computing each of the nine cases here that the terms in this product have a general structure of the form

$$\left( P_j^{\alpha_j} \right)^\dagger P_j^{\beta_j} = \delta_{\alpha_j, \beta_j} + \text{const} \times P^\pm. \tag{38}$$

Therefore, we may expand the product in Eq. 37 to produce a sum of vacuum expectations of strings of $P$-operators. However, any site acted on by a single $P^\pm$ operator will be wound away from $|\tilde{0}\rangle$ with probability 1. Therefore, any string containing a $P^+$ or $P^-$ must have a zero vacuum expectation and $\langle \boldsymbol{\alpha} | \boldsymbol{\beta} \rangle = \delta_{\boldsymbol{\alpha}, \boldsymbol{\beta}}$ as required.

To prove that the parafermion braiding matrices reproduce the $SU(2)_4$ braiding matrices we will now verify the equivalence for all possible cases. I.e., we should demonstrate that our parafermionic representation of the braid matrices reproduces every relation in table 1.

**Even cases:** $B_{2j}$ — We start by considering the even braid matrices $B_{2j}$. In terms of the $P^+$ operators we can write

$$B_{2j} = \frac{\alpha \omega}{\sqrt{3}} \left( \mathbb{1}_j + \omega^4 \sqrt{2} P_j^+ \right). \tag{39}$$

Thus the braid matrix here acts only on the $j^{th}$ bond.

Note that in the language of the Bratteli diagram, $B_{2j}$ alters a path that moves from a half-integer spin height to an integer spin height and back to a half-integer spin height. These are all the paths shown in the first five lines of table 1.

There are three possibilities of states that the matrix may operate on — $\mathbb{1}_j$, $P_j^+$, or $P_j^-$ applied to $|\Phi_N\rangle$, corresponding to the kets in Eqs. 32-34. We will examine the action of the braid operator applied to each of these states.

Acting on $\mathbb{1}_j |\Phi_N\rangle = |\Phi_N\rangle$ (Eq. 32), the result is of the form

$$B_{2j} |\Phi_N\rangle = \frac{\alpha \omega}{\sqrt{3}} |\Phi_N\rangle - \frac{\alpha \sqrt{2}}{\omega \sqrt{3}} P_j^+ |\Phi_N\rangle. \tag{40}$$

Identifying the kets on the right hand side by using Eq. 33, we see that we have proven the first and second equations in table 1.

Let us now consider $B_{2j}$ instead acts on $P_j^+ |\Phi_N\rangle$, i.e., we are considering acting on Eq. 33. Here we must use the identity

$$\left( P_j^+ \right)^2 = \frac{P_j^+}{\sqrt{2}} + \mathbb{1}_j \tag{41}$$

which is easily proved by direct calculation. Now applying $B_{2j}$ in the form of Eq. 39 to the ket $P_j^+ |\Phi_N\rangle$ and using the above identity we find that

$$B_{2j} P_j^+ |\Phi_N\rangle = \frac{\alpha \omega^3}{\sqrt{3}} P_j^+ |\Phi_N\rangle - \frac{\alpha \sqrt{2}}{\omega \sqrt{3}} |\Phi_N\rangle. \tag{42}$$

This similarly proves the third and fourth identities in table 1.

Finally, we can act the braid matrix on $P_j^-$. Using the identity

$$P_j^+ P_j^- = -\frac{P_j^-}{\sqrt{2}} \tag{43}$$

we similarly obtain

$$B_{2j} P_j^- |\Phi_N\rangle = \alpha P_j^- |\Phi_N\rangle.$$

which then establishes the fifth line of table 1.

**Odd cases:** $B_{2j-1}$ — We will now consider the odd braiding matrices, $B_{2j-1}$. These braid matrices act on Bratteli diagrams that go from integer-spin to half-integer-spin and back to integer-spin, as shown in the final five lines of table 1.

The action of $B_{2j-1}$ is extremely simple. From the structure of the braid matrix in Eq. 27, we notice that this operator serves only to measure $\tau_j$ on one clock site with the results

$$B_{2j-1} |\tilde{0}\rangle_j = \alpha \omega^4 |\tilde{0}\rangle_j, \tag{44}$$

$$B_{2j-1} |\tilde{1}\rangle_j = \alpha |\tilde{1}\rangle_j, \qquad B_{2j-1} |\tilde{2}\rangle_j = \alpha |\tilde{2}\rangle_j. \tag{45}$$

When considering a Bratteli state on which $B_{2j-1}$ acts, as usual we start with $|\Phi_N\rangle$ then allow the action of $P^\pm$ on each clock site. The $j^{\text{th}}$ clock site can be acted upon by $P_j^\pm$ or $\mathbb{1}_j$ from the right bond and also by $P_{j-1}^\pm$ or $\mathbb{1}_{j-1}$ from the left. In those cases where both operators on right and left are the identity, the state simply picks up a phase $\alpha \omega^4$ as $B_{2j-1}$ acts only on $|\tilde{0}\rangle_j$ states. This proves line 6 of table 1. On the other hand, in the cases where one of the bond operators is the identity and the other is some $P^\pm$ the clock at site $j$ is wound to be some superposition of only $|\tilde{1}\rangle_j$ and $|\tilde{2}\rangle_j$. Therefore, acting with $B_{2j-1}$ simply produces the phase $\alpha$, proving lines 7 and 8 of table 1.

There is, however, one more complicated case. $P_{j-1}^\pm P_j^\pm$ winds the clock on site $j$ twice, with some amplitude of returning it to $|\tilde{0}\rangle_j$. Explicitly one finds that

$$P_{j-1}^\mu P_j^\nu |\Phi_N\rangle = \frac{1}{2} \left( \sigma_{j-1}^\dagger \sigma_{j+1} + \mu\nu \sigma_{j-1} \sigma_{j+1}^\dagger \right) |\Phi_N\rangle$$
$$+ \frac{1}{2} \left( \mu \sigma_{j-1} \sigma_j \sigma_{j+1} + \nu \sigma_{j-1}^\dagger \sigma_j^\dagger \sigma_{j+1}^\dagger \right) |\Phi_N\rangle, \tag{46}$$

where $\mu$ and $\nu$ are $+$ or $-$ (but not necessarily equal). In the first term of this equation the site $j$ has been wound back to $|\tilde{0}\rangle_j$, and so acting with $B_{2j-1}$ returns a factor of $\alpha \omega^4$. However, in the second term the site $j$ is in a superposition of $|\tilde{1}\rangle_j$ and $|\tilde{2}\rangle_j$, so $B_{2j-1}$ returns $\alpha$. Explicitly,

$$B_{2j-1} P_{j-1}^\mu P_j^\nu |\Phi_N\rangle = \frac{\alpha \omega^4}{2} \left( \sigma_{j-1}^\dagger \sigma_{j+1} + \mu\nu \sigma_{j-1} \sigma_{j+1}^\dagger \right) |\Phi_N\rangle$$
$$+ \frac{\alpha}{2} \left( \mu \sigma_{j-1} \sigma_j \sigma_{j+1} + \nu \sigma_{j-1}^\dagger \sigma_j^\dagger \sigma_{j+1}^\dagger \right) |\Phi_N\rangle, \tag{47}$$

Therefore, $B_{2j-1}$ mixes the state $P_{j-1}^\mu P_j^\nu |\Phi_N\rangle$ with another state in which these same two terms appear, but with different coefficients. The only other state which includes both these terms is $P_{j-1}^{-\mu} P_j^{-\nu} |\Phi_N\rangle$ and so, in general these two are mixed. It is simple to check that the correct combination which reproduces the right hand side of Eq. 47 is then

$$B_{2j-1} \left( P_{j-1}^\mu P_j^\nu |\Phi_N\rangle \right) = \frac{\alpha \omega^2}{2} \left( P_{j-1}^\mu P_j^\nu |\Phi_N\rangle \right) - \frac{\alpha\sqrt{3}}{2\omega} \left( P_{j-1}^{-\mu} P_j^{-\nu} |\Phi_N\rangle \right). \tag{48}$$

This reproduces the final two braiding relations of table 1, completing our proof.

## 5  Conclusions

We have shown that the braid group representation for SU(2)$_4$ anyons can be described by $\mathbb{Z}_3$ parafermionic operators as given by Eqs. 27 and 28, with the Hilbert space replicated by the operators in Eqs. 32-34. This result is exciting for a number of reasons. First, this structure has the potential to greatly simplify calculations relating to SU(2)$_4$ anyons by reducing computations to simple linear algebra with basic operators. Secondly, this work further reinforces the deep links between SU(2)$_4$ and $\mathbb{Z}_3$ parafermions and may result in new ideas for how computationally universal SU(2)$_4$ anyons might be realised in practice.

## Acknowledgements

We are grateful to J. Slingerland, R. Bondesan, K. Shtengel, and E. Berg for enlightening discussions. This work was supported by EPSRC grants EP/I031014/1 and EP/N01930X/1. Statement of compliance with EPSRC policy framework on research data: This publication is theoretical work that does not require supporting research data.

## A  Verification of braid relations

Here we check that our braid matrices (Eq. 26 or equivalently Eqs. 27 and 28) form a braid group representation, in that they satisfy the relations

$$B_j B_{j+m} = B_{j+m} B_j, \tag{49}$$

$$B_j B_{j+1} B_j = B_{j+1} B_j B_{j+1}, \tag{50}$$

for $m \geq 2$. The first of these relations is simple to confirm by inspection as the operators $\sigma$ and $\tau$ commute unless on the same site, which is excluded by the $m \geq 2$ condition. The latter relation is more complicated to prove but, upon rewriting all the matrices as some $B_j = \frac{\alpha\omega}{\sqrt{3}}\left(1 + \omega^4 Y_j\right)$ where $Y_j = x_j + x_j^\dagger$ for some $x_j$ satisfying $x_j^\dagger x_j = x_j x_j^\dagger = 1$, this relation reduces to

$$Y_j Y_{j+1} Y_j - Y_j = Y_{j+1} Y_j Y_{j+1} - Y_{j+1} \tag{51}$$

(where we have used that $Y_j^2 = 2 + Y_j$ for the conditions given). It is then simple to check that these $x_j$ satisfy $x_j x_{j+1} = \lambda x_{j+1} x_j$ and so the left hand side of Eq. 51 can be written as

$$Y_j Y_{j+1} Y_j - Y_j = x_j x_{j+1} x_j + x_{j+1} x_j x_{j+1} + x_j x_{j+1}^\dagger x_j + x_j^\dagger x_{j+1} x_j^\dagger - Y_{j+1} - Y_j. \tag{52}$$

This expression must be unchanged if we swap $j$ with $j+1$. This is clearly true for the first two and last two terms and is easy to confirm for the middle two using only the algebra of the $x$ operators stated above. Thus, our proposed matrices do indeed satisfy both braiding relations.

## B  Motivation for the form of the braid matrix ansatz

In the main text our strategy was to simply claim the connection between the SU(2)$_4$ anyons and the $\mathbb{Z}_3$ parafermions and then prove it is true. This may seem a bit mysterious. In this appendix we show how certain constraints in SU(2)$_4$ leads one inevitably to this particular relation. We also elaborate on what additional choice we have in defining an alternate mapping between SU(2)$_4$ anyons and the $\mathbb{Z}_3$ parafermions.

**1) Dimensionality** - We may construct the Hilbert space and braiding matrices with only simple considerations about the braiding relations for SU(2)$_4$. We begin by postulating that every two-particle path segment on the Bratteli diagram corresponds to one site of a $\mathbb{Z}_3$ clock chain such that the numbers of SU(2)$_4$ anyons equals the number of $\mathbb{Z}_3$ parafermions. To get the correct dimensionality then the path through this space must be described by operators living on the bonds of this clock chain, of which there are three per bond.

**2) Translational invariance** - We then require some vacuum state upon which to build our Hilbert space. The natural choice is the state which represents $\left|\overline{\overline{\wedge\wedge\wedge}}\right\rangle$. This is the only state which is 'translationally' invariant (i.e., the braiding relations repeat every two particles). Therefore, this is also the only state which must be represented by a translationally invariant clock state.

Of course, no choice of translationally invariant clock state is unique as each clock could be set to $|0\rangle$, $|1\rangle$, $|2\rangle$, or any combination thereof. Three such combinations are the states for which $\tau$ is a good quantum number, defined by $\tau|\tilde{l}\rangle = \lambda^l|\tilde{l}\rangle$. In the main text we called this the Fourier basis. Thankfully, our choice of basis is of little importance as alternative formulations can be reached by redefining our operators $\sigma$ and $\tau$. For example, if we were to choose $|0\rangle$ on each site then we could easily switch to $|1\rangle$ by redefining $\sigma \to \lambda^{-1}\sigma$. Furthermore, if we wish to switch to the $\tau$ basis then the conversion is achieved by replacing $\sigma$ and $\tau$ with $\sigma' = \tau$ and $\tau' = \sigma^\dagger$. In the main text we have chosen to work in the basis where $\tau$ is a good quantum number as this will make the braid matrices local in the parafermions, so

$$\left|\overline{\overline{\wedge\wedge\wedge}}\right\rangle \equiv |\Phi_N\rangle := \left|\ldots,\tilde{0},\tilde{0},\tilde{0},\ldots\right\rangle. \tag{53}$$

In this basis we can think of the value of $\tau$ as the direction in which the clock points and $\sigma$ as the winding operator, $\sigma|\tilde{l}\rangle = \widetilde{|l-1\rangle}$.

**3) Odd braids** - For the state 53 the action of the odd braiding matrices, $\mathscr{B}_{2j-1}$, must produce a phase $\alpha\omega^4$. Therefore, the braiding matrix in the parafermionic language, $B_{2j-1}$, must be an eigenoperator of $|\Phi_N\rangle$, so must be constructed purely from $\tau$ operators. Furthermore, it must be unaware of the states of neighbouring clocks as it acts only on the two-path segment corresponding to the $j^{\text{th}}$ clock site and therefore depends only on $\tau_j$. Thus, in general

$$B_{2j-1} = a + b\tau_j + c\tau_j^\dagger. \tag{54}$$

Then, given that the braid relation explicitly reads

$$\mathscr{B}_{2j-1}\left|\overline{\overline{\wedge}}\right\rangle = \alpha\omega^4\left|\overline{\wedge}\right\rangle \tag{55}$$

we have our first constraint that $a + b + c = \alpha\omega^4$.

**4) Even braids** - If we braid anyon $2j$ with anyon $2j+1$ in this $|\Phi_N\rangle$ state then the result is of the form

$$\mathscr{B}_{2j}\left|\overline{\overline{\smile}}\right\rangle = \frac{\alpha\omega}{\sqrt{3}}\left|\overline{\overline{\smile}}\right\rangle - \frac{\alpha\sqrt{2}}{\omega\sqrt{3}}\left|\overline{\wedge}\right\rangle. \tag{56}$$

This generates the new state $\left|\overline{\wedge}\right\rangle$ which transforms as follows under the same braiding matrix $\mathscr{B}_{2j}$,

$$\mathscr{B}_{2j}\left|\overline{\wedge}\right\rangle = \frac{\alpha\omega^3}{\sqrt{3}}\left|\overline{\wedge}\right\rangle - \frac{\alpha\sqrt{2}}{\omega\sqrt{3}}\left|\overline{\overline{\smile}}\right\rangle. \tag{57}$$

In what follows we will also use the fact that the SU(2)$_4$ braiding matrices satisfy $\mathscr{B}_{2j}^3 = \alpha^3$.

To replicate the first relation, Eq. 56 we propose that the even braiding matrices in the language of the clock model are of the form

$$B_{2j} = \frac{\alpha\omega}{\sqrt{3}}\left(1 + \omega^4\sqrt{2}P_j^+\right) \tag{58}$$

where this $P_j^+$ is some operator which generates a state orthogonal to $|\Phi_N\rangle$ and represents the $\left|\overline{\smile}\right\rangle$ path segment, i.e.,

$$P_j^+ |\Phi_N\rangle \equiv \left|\overline{\wedge\wedge\wedge}\right\rangle. \tag{59}$$

To satisfy the second relation, Eq. 57, and ensure that $B_j^3 = \alpha^3$ we conclude respectively that

$$\left(P_j^+\right)^2 = 1 + \frac{P_j^+}{\sqrt{2}}, \qquad \left(P_j^+\right)^\dagger = P_j^+. \tag{60}$$

Finally, we may consider that the action of $P_j^+$ can only change the site $|\tilde{l}\rangle$ on site $j$ and $j+1$ and therefore can depend only on $\tau_j, \sigma_j, \tau_{j+1}$ and $\sigma_{j+1}$. However, including $\tau_j$ in the definition is unnecessary as it can be commuted through the $\sigma_j$ operators to act on $|\Phi_N\rangle$ and therefore simply produce a phase. Thus

$$P_j^+ = P_j^+(\sigma_j, \sigma_{j+1}). \tag{61}$$

**5) Consistency** - Ensuring that the two ansätze are consistent will fix all coefficients in both braiding matrices. Firstly, we note that

$$\mathscr{B}_{2j-1} \left|\overline{\diagup}\right\rangle = \alpha \left|\overline{\diagup}\right\rangle, \qquad \mathscr{B}_{2j+1} \left|\overline{\diagdown}\right\rangle = \alpha \left|\overline{\diagdown}\right\rangle. \tag{62}$$

Therefore, it must be the case that $P_j^+ |\Phi_N\rangle$ is an eigenstate of $B_{2j-1}$ and $B_{2j+1}$ with eigenvalue $\alpha$. This forces two conclusions. Firstly, because we claim $B_{2j-1}|\tilde{0}\rangle = \alpha\omega^4|\tilde{0}\rangle$, it must be the case that neither clock $j$ nor $j+1$ are in the $|\tilde{0}\rangle$ state. Therefore, $P_j^+$ must wind both clocks $j$ and $j+1$ away from zero, and so can only be made from four terms, namely $\sigma_j\sigma_{j+1}, \sigma_j\sigma_{j+1}^\dagger, \sigma_j^\dagger\sigma_{j+1}$ and $\sigma_j^\dagger\sigma_{j+1}^\dagger$. Given the identities for $P_j^+$ in Eq. 60 we must then conclude that either

$$P_j^+ = \frac{1}{\sqrt{2}}\left(\sigma_j\sigma_{j+1}^\dagger + \sigma_j^\dagger\sigma_{j+1}\right)$$
$$\text{or} \quad P_j^+ = \frac{1}{\sqrt{2}}\left(\sigma_j\sigma_{j+1} + \sigma_j^\dagger\sigma_{j+1}^\dagger\right). \tag{63}$$

In the main text we use the former case.

This revelation also constrains the odd braiding matrix as it imposes two new constraints. Given that we must have $B_{2j-1}|\tilde{1}\rangle = \alpha|\tilde{1}\rangle$ and $B_{2j-1}|\tilde{2}\rangle = \alpha|\tilde{2}\rangle$ we realise that

$$a + b\lambda + c\lambda^2 = a + b\lambda^2 + c\lambda = \alpha \tag{64}$$

where $a, b$ and $c$ were the coefficients in $B_{2j-1}$ defined in Eq. 54. Therefore, we find that

$$B_{2i-1} = \frac{\alpha\omega}{\sqrt{3}}\left(1 + \omega^4\left(\tau_j + \tau_j^\dagger\right)\right). \tag{65}$$

**6) Completing the Hilbert space** - So far we have constrained the form of both even and odd braiding matrices in equations 58 and 65. Furthermore, we have found two of the three states in our Hilbert space, namely

$$\mathbb{1}_j |\Phi_N\rangle \equiv \left|\overline{\smile}\right\rangle \qquad \text{and} \qquad P_j^+ |\Phi_N\rangle \equiv \left|\overline{\frown}\right\rangle. \tag{66}$$

The third and final allowable path segment is $P_j^- |\Phi_N\rangle \equiv \left| \diagup \right\rangle$, where this new operator $P_j^-$ defines a state which must be orthogonal to both $P_j^+ |\Phi_N\rangle$ and $|\Phi_N\rangle$. Therefore, we conclude that

$$P_j^- = e^{i\phi} \frac{1}{\sqrt{2}} \left( \sigma_j \sigma_{j+1}^\dagger - \sigma_j^\dagger \sigma_{j+1} \right)$$

$$\text{or} \quad P_j^- = e^{i\phi} \frac{1}{\sqrt{2}} \left( \sigma_j \sigma_{j+1} - \sigma_j^\dagger \sigma_{j+1}^\dagger \right) \tag{67}$$

for some a priori arbitrary phase $\phi$. However, by considering the action of $B_{2j-1}$ on pairs of path operators, for example the state $P_{j-1}^+ P_j^+ |\Phi_N\rangle \equiv \left| \frown\frown \right\rangle$, and comparing with the expected braiding relations for SU(2)$_4$ we find that this phase must be $e^{i\phi} = \pm 1$. We take the $e^{i\phi} = +1$ solution and the first case of Eq. 67 for the calculations in the main text.

**Appendix B summary** - We were able to derive the forms of the states in the Hilbert space and the forms of the braiding matrices under very general considerations. In this way the majority of the braiding relations of SU(2)$_4$ are replicated automatically by this construction. It is simple to check then, as discussed in the main text, that the remaining braiding relations are also consistent with this picture.

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
