# Peer review of "How SU(2)$_4$ Anyons are Z$_3$ Parafermions"

_SciPost Physics, doi:SciPost Phys. 3, 037 (2017)_

## Round 1 · Referee Report · Anonymous (Referee 1) · 2017-8-3

Strengths

  1. explicitly establishing a detailed relation at the level of braid group representation between the braiding statistics of SU(2)_4 anyons and the algebra of Z_3 parafermionic operators

Weaknesses

  1. the use of terminology in this paper is confusing and potentially misleading

Report

The value of this paper is in explicitly establishing a detailed relation at the level of braid group representation between the braiding statistics of SU(2)_4 anyons and the algebra of Z_3 parafermionic operators. I think this work deserves to be published.

However, I consider the use of terminology in this paper to be confusing and potentially misleading and think it is necessary to correct this before publication. Throughout this paper, the authors should make it unambiguously clear when they are referring to parafermionic operators, which are mathematical objects that act in Hilbert space, as opposed to “parafermions,” which is a somewhat ambiguous term, but usually refers to physical objects that are (quasi)particle-like with root of unity exchange statistics (i.e. Abelian anyons), and both of these should also be clearly distinguished from parafermion zero modes, which are physical objects that are defects (into which parafermions can tunnel without changing the energy of the defect) with non-Abelian exchange statistics. It is, of course, the case that parafermionic operators can be used to provide a representation of the braiding statistics of parafermion zero modes, but one should not equate the two and, in fact, the authors specifically are utilizing the mathematical objects (parafermionic operators) with no concern to the notions of stability or physical realization associated with the parafermionic zero mode defects. (This is similar to the situation with Majorana fermionic operators, Majorana fermions, and Majorana zero modes.) This problem is highlighted by the title “How SU(2)$_4$ Anyons are Z$_3$ Parafermions,“ which is a bad title seeing as how it is incorrect to say these two things are the same, for any usage/misusage of the term “Parafermions.” A better title would be something along the lines of “A braid group representation of SU(2)$_4$ Anyons from the algebra of Z$_3$ Parafermion Operators.”

I would also like to point out that the relationship between SU(2)_4 Anyons and Z$_3$ Parafermion zero modes (which, as noted above, is related to the algebra of parafermionic operators) has previously been made much more concrete than suggested in the introduction and in any of the cited references. Specifically, the paper arXiv:1410.4540 has fully described the algebraic theory of symmetry defects in topological phases (which describes Z_3 parafermion zero modes) in terms of G-crossed category theory and the relation between the defect theory and the theory obtained by gauging the symmetry (which describes SU(2)_4 anyons) has been explained in detail. Moreover, the basic data has been worked out in this paper for the example of Z_N parafermionic zero modes, and their relation to SO(N)_2 anyons noted, making the relation between the braiding statistics of these theories fairly explicit (much more so than previous references, though somewhat less explicitly in terms of braid group representations than what has been worked out in the authors’ paper).

Requested changes

See report

---

## Round 1 · Referee Report · Anonymous (Referee 2) · 2017-8-21

Strengths

Very explicit and physical illustration of a general theory and accessible to physicists.

Weaknesses

Some standard confusion as to the difference between anyons and their various different avatars.

Report

The paper is a very interesting and useful addition to the literature on a very interesting generalization of Majoranas to parafermions. I recommend its publications with the following requested changes.

Requested changes

  1. Clarify, as already pointed out, the differences between anyons and defects.
  2. Add the reference of Jones, V. Jones, Braid groups, Hecke algebras and type II1 factors, in Geometric Methods in Operator Algebras (Kyoto, 1983), Pitman Res. Notes Math. Ser. 123, Longman Sci. Tech., Harlow, 1986, 242–273. Harlow, 1986, where the main result appeared in a slightly different context.
  3. The citation of 14 for universality with additional operations is inaccurate in two aspects: a) ref 14 discussed a slightly different theory--the Jones version, b) the universality for SU(2)_4 is first proved in a different reference arXiv:1405.7778 Universal Quantum Computation with Metaplectic Anyons, Shawn X. Cui, Zhenghan Wang

---

## Round 1 · Referee Report · Anonymous (Referee 3) · 2017-8-23

Strengths

1 Interesting, well written, to the point paper

Weaknesses

1 Only minor issues, see report

Report

Report on 'How $su(2)_4$ anyons are $Z_3$ parafermions'.

This paper discusses the relation between the anyons of
$su(2)_4$ and $Z_3$ parafermionic zero-modes in detail, in
particular for the braid group representation of the
l=1/2 representation of $su(2)_4$.

The authors start by reviewing the braid group representation
associated with n l=1/2 anyons of $su(2)_4$, giving the explicit
action of the braid generators on the Hilbertspace of n l=1/2
anyons. To make the connection with the $Z_3$ parafermionic
zero modes, the authors show explicitly that these braid
generators can be constructed in terms of the $Z_3$ parafermions
(or, equivalently, in terms of the 'clock basis').

As the authors state, establishing this relation is an important
result, because it gives a 'simple' operator algebra 'implementation'
of the $su(2)_4$ non-abelian anyons.

The paper is well written, and for the most easy to follow. One point
which is slightly unclear concerns the Hilbertspace in the clock basis,
and 'equations' (32-34). Here, it seems that which alternative is
relevant, depends on which operators are chosen at the bonds with
labels not equal to j. Is it true that all of these states are by construction
orthonormal, or are they declared to be orthonormal? This issue is
related to a question I have about the proof of the odd cases, starting
on page 5. In the more complicated case, it seems the authors are merely
matching the coefficients a and b in eq. 44, while it should be possible
to derive them explicitly by evaluating the lhs of eq. 44. In any case,
to establish the result, one should show that the normalization of the two
states in the 'clock Hilbertspace' are the same, and independent of the
choices of the operators with labels different from j-1 and j (even though
this seems simple to show, I think it should be mentioned).

I appreciated app. B, in which the authors construct the result, which in
the main text is posed, and checked for consistency.

Some final comments/questions.

The construction seems to hinge on the fact that the quantum dimension of
the l=1/2 anyon is a simple root. This is not the case for the l=1/2 anyon
in $su(2)_{10}$ (which has central charge c=5/2), even though they take a simple
form. Does this exclude a similar construction for $su(2)_{10}$?

I suggest, for obvious reasons, that the authors swap reference one and two.

In conclusion, this is an interesting, and well written paper. I recommend it
for publication in SciPost, but ask the authors to consider the points I brought
up.

Requested changes

See report

---

## Round 2 · Referee Report · Anonymous (Referee 3) · 2017-11-7

Report
2nd Report on 'How su(2)_4 anyons are Z_3 parafermions'.
The authors took the reports of the three referees as an
opportunity to improve the manuscript. They took the suggestions
into account, and in my opinion, they have been clear with the
distinction between su(2)_4 anyons and Z_k parafermion operators
a la Fradkin-Kadanoff, and I agree with the authors that the
chosen title is appropriate. I recommend publication of the
paper in SciPost in its current from.
The authors took the reports of the three referees as an
opportunity to improve the manuscript. They took the suggestions
into account, and in my opinion, they have been clear with the
distinction between su(2)_4 anyons and Z_k parafermion operators
a la Fradkin-Kadanoff, and I agree with the authors that the
chosen title is appropriate. I recommend publication of the
paper in SciPost in its current from.

---

## Round 2 · Author Response

We thank the referees for their comments. All of the referee comments are well thought out and we agree with the suggested changes. We have addressed all of the comments of the referees, and we believe that our manuscript now is suitable for publication in SciPost. Detailed responses to the referee comments are given below along with description of the changes.
Referee 1:
We thank the referee for their careful reading and their comments.
(1) We have extended the discussion near Eqs. 32-34 to clarify the issue, in particular we also show how the states are orthonormal (and we state this explicitly). In addition near eqn 44 (now eqn 48) we have simplified the argument so that it is hopefully now clearer (our previous discussion of fitting parameters a and b was perhaps needlessly awkward).
(2) We have added a paragraph (Between Eq. 35 and 36) commenting on the fact that among theories SU(2)_k only k=1,2,4 could possibly be represented with the parafermion algebra. It is true that a few other cases seem to be simpler than the generic case (and might have some simplifying structure), however, they certainly will not fit a similar parafermion construction.
(3) We have changed the order of Refs 1 and 2 .
Referee 2:
We thank the referee for their careful reading and their comments.
(1) The confusion about 'avatars' of parafermions was also mentioned by Referee 3. We have attempted to straighten this out by being much more explicit about what we mean mathematically. See below the response to Referee 3 below for more details.
(2) We thank the referee for pointing us to this beautiful reference by Jones which we have now cited. In addition we have added several additional references to other related work (for example, that of Saleur 1991; and a number of recent related works).
(3) We have fixed the reference problem and added the citation to Cui and Zhang. We have also kept the arXiv number so that the order of appearance in the literature is clear.
Referee 3:
We thank the referee (Parsa Bonderson) for his careful reading and his comments.
(1) Parsa is certainly right that we were not careful about specifying exactly what we mean by "parafermion". This is particularly bad being that the literature is not very consistent in how this word is used. We have thus gone through the paper and changed some of the language to clarify precisely what we mean and exactly how we have used this word. We believe these changes should remove any confusion. Details of the changes are listed below.
(2) Parsa also points to his paper arXiv:1410.4540 as having worked out details of how ``parafermions" arise as symmetry defects of topological phases. This is certainly very closely related. We emphasize that we are agnostic about the physical mechanism for the production of these objects (indeed, defect parafermions have projective statistics, so are not quite the same as what we are considering). While we have added citations to his work, along with some additional works in the literature working along similar lines, we do not add a detailed discussion of the physics of symmetry defects since it is a bit off-topic to our main focus.
Explict changes regarding point (1):
We have explicitly stated exactly what we mean by parafermion with the addition "To avoid confusion we emphasize at this point that within this paper, all mention of parafermions will refer to Fradkin-Kadanoff-Fendley type --- that is, they are mathematical operators (to be defined precisely in Eqs. 19 and 20 below)". We also have removed a number of statements that were potentially creating confusion. Statements such as "Thus, we have proven the equivalence between SU(2)4 anyons and Z3 parafermions" (was in the conclusion now removed) which we agree is not precise and probably adds confusion. We also rephrase the key sentence in the introduction that explains the purpose of the paper to now read as: "The content of the current work is to express the braid group representation corresponding to SU(2)4 anyons explicitly in terms of Z3 parafermion operators". We believe these changes should make our usages unambiguous.
One thing we did not change is the title, "How SU(2)4 anyons are Z3 parafermions" as we still believe that this is accurate, despite the objections. If one is to discuss "How A is B" it often implies that "A is not always B", or "A is B only in some sense", or "A is a subclass of B". If our aim was to state that they are the same then "A is B", or "How A is exactly B" would probably be more appropriate. Therefore, while it is true that the title suggested by Parsa is also accurate, it is also somewhat long and unwieldy. Furthermore, changing our title between the first and second draft of a paper is likley to add confusion. So unless forced, we would prefer to stick with the original title.
Referee 1:
We thank the referee for their careful reading and their comments.
(1) We have extended the discussion near Eqs. 32-34 to clarify the issue, in particular we also show how the states are orthonormal (and we state this explicitly). In addition near eqn 44 (now eqn 48) we have simplified the argument so that it is hopefully now clearer (our previous discussion of fitting parameters a and b was perhaps needlessly awkward).
(2) We have added a paragraph (Between Eq. 35 and 36) commenting on the fact that among theories SU(2)_k only k=1,2,4 could possibly be represented with the parafermion algebra. It is true that a few other cases seem to be simpler than the generic case (and might have some simplifying structure), however, they certainly will not fit a similar parafermion construction.
(3) We have changed the order of Refs 1 and 2 .
Referee 2:
We thank the referee for their careful reading and their comments.
(1) The confusion about 'avatars' of parafermions was also mentioned by Referee 3. We have attempted to straighten this out by being much more explicit about what we mean mathematically. See below the response to Referee 3 below for more details.
(2) We thank the referee for pointing us to this beautiful reference by Jones which we have now cited. In addition we have added several additional references to other related work (for example, that of Saleur 1991; and a number of recent related works).
(3) We have fixed the reference problem and added the citation to Cui and Zhang. We have also kept the arXiv number so that the order of appearance in the literature is clear.
Referee 3:
We thank the referee (Parsa Bonderson) for his careful reading and his comments.
(1) Parsa is certainly right that we were not careful about specifying exactly what we mean by "parafermion". This is particularly bad being that the literature is not very consistent in how this word is used. We have thus gone through the paper and changed some of the language to clarify precisely what we mean and exactly how we have used this word. We believe these changes should remove any confusion. Details of the changes are listed below.
(2) Parsa also points to his paper arXiv:1410.4540 as having worked out details of how ``parafermions" arise as symmetry defects of topological phases. This is certainly very closely related. We emphasize that we are agnostic about the physical mechanism for the production of these objects (indeed, defect parafermions have projective statistics, so are not quite the same as what we are considering). While we have added citations to his work, along with some additional works in the literature working along similar lines, we do not add a detailed discussion of the physics of symmetry defects since it is a bit off-topic to our main focus.
Explict changes regarding point (1):
We have explicitly stated exactly what we mean by parafermion with the addition "To avoid confusion we emphasize at this point that within this paper, all mention of parafermions will refer to Fradkin-Kadanoff-Fendley type --- that is, they are mathematical operators (to be defined precisely in Eqs. 19 and 20 below)". We also have removed a number of statements that were potentially creating confusion. Statements such as "Thus, we have proven the equivalence between SU(2)4 anyons and Z3 parafermions" (was in the conclusion now removed) which we agree is not precise and probably adds confusion. We also rephrase the key sentence in the introduction that explains the purpose of the paper to now read as: "The content of the current work is to express the braid group representation corresponding to SU(2)4 anyons explicitly in terms of Z3 parafermion operators". We believe these changes should make our usages unambiguous.
One thing we did not change is the title, "How SU(2)4 anyons are Z3 parafermions" as we still believe that this is accurate, despite the objections. If one is to discuss "How A is B" it often implies that "A is not always B", or "A is B only in some sense", or "A is a subclass of B". If our aim was to state that they are the same then "A is B", or "How A is exactly B" would probably be more appropriate. Therefore, while it is true that the title suggested by Parsa is also accurate, it is also somewhat long and unwieldy. Furthermore, changing our title between the first and second draft of a paper is likley to add confusion. So unless forced, we would prefer to stick with the original title.

---

## Round 2 · List of Changes

Page 1 - Changed the wording of the abstract to be more careful with regards to parafermion operators
Page 1 - Reworded a number of parts throughout the introduction and added extras which we hope should make our usage of the relevant terminology clearer
Page 4 - Added a comment to explain the "or" in Eqs. 32-34
Page 5 - Added a proof that our states are orthonormal
Page 6 - Reworked the final claculation of the proof, which hopefully clarifies matters
New references are [14], [16], [30], [31], [36], [37], [38], [39].
Page 1 - Reworded a number of parts throughout the introduction and added extras which we hope should make our usage of the relevant terminology clearer
Page 4 - Added a comment to explain the "or" in Eqs. 32-34
Page 5 - Added a proof that our states are orthonormal
Page 6 - Reworked the final claculation of the proof, which hopefully clarifies matters
New references are [14], [16], [30], [31], [36], [37], [38], [39].

---

## Editorial Decision

published